# Nurses’ Participation in the Psychiatric Recovery Process: A Qualitative Study in Psychiatric Intensive Care Units in Chile

**DOI:** 10.3390/nursrep15110391

**Published:** 2025-11-06

**Authors:** Daniela Fuentes-Olavarría

**Affiliations:** Nursing School, Faculty of Medicine, Clínica Alemana de Santiago, Universidad del Desarrollo, Santiago 7610315, Chile; dfuentes@udd.cl; Tel.: +56-2-23279310

**Keywords:** mental health recovery, psychiatric nursing, psychiatric hospitals, inpatient care units, qualitative research

## Abstract

**Background**: Recovery is an emerging approach. In Chile, attempts are being made to introduce the Recovery Model with specific guidelines for the care of people diagnosed with Severe Mental Disorders. The participation of nurses in this process is peripheral to the biomedical model. **Objectives**: To explore the participation of nurses in the recovery process of people hospitalised in Psychiatric Intensive Care between 2023 and 2024. **Methods**: Qualitative research, collective-case multisite study design in four hospitals. With the approval of four ethics committees, 18 nurses who signed informed consent were interviewed. Rapid qualitative analysis was performed. **Results**: Nursing care is mainly related to the caregiving, educational, and management roles. Recovery is associated with clinical improvement, and different components are identified, such as family and social support, the ability to resume control of one’s life, the existence of a future life plan, and the ability to manage one’s own illness. **Conclusions**: The results are consistent with elements described in contemporary approaches to recovery, incorporating autonomy, confidence in the person’s abilities, and shared decision-making. However, they are still far from modern approaches to personal and non-clinical recovery. Nursing needs to redirect its efforts toward recovery with a paradigm shift toward a model in which the person affected by a mental health condition is the protagonist of their own health process.

## 1. Introduction

Unlike developed countries, recovery is an emerging approach in Chile. Since the Psychiatric Reform in Europe, there has been a shift towards a Community Mental Health Model [1] that respects human rights and protects people with mental health or psychiatric conditions [2], a change that has been reflected in the National Mental Health Plan [3]. Convinced that people with mental health problems can recover and live in the community [4], the Chilean Ministry of Health has published recommendations for mental health professionals on the Recovery Model. These new guidelines, specific to the recovery of people diagnosed with Severe Mental Disorders (SMDs) [5], have been developed because traditional approaches in institutional settings do not recognise the importance of encouraging participation in decisions about personal therapeutic processes [6]. Although a large number of people recover from SMD, it has been reported that up to 20% will continue to experience serious and complex long-term consequences [4]. In many of these cases, the continuous violation of their rights results in low adherence, poor outcomes [7], rehospitalisations [8], increased care costs [9], prolonged treatments, low family participation [10], and reduced life expectancy [11].

Recovery is a complex construct that includes various components for which different conceptual frameworks have been proposed [12]. Thanks to the early work of Corrigan, there has been a shift from the concept of clinical recovery, traditionally defined by mental health professionals, to a definition of personal recovery in mental health. The new approaches consider recovery as a process through which people try to achieve goals in life, in a context of hope, despite the symptoms they experience [13], in which cultural norms, social expectations and community interactions come together [14]. It involves the possibility of building a life beyond illness, without necessarily achieving the elimination of symptoms, with goals that the individuals themselves define [15]. It is, therefore, a unique process of changing attitudes, values, feelings, goals, skills, and roles within society, which allows people to live a satisfying life, with hope and full of contributions, even with the limitations caused by the illness [16]. Recovery involves the development of new meaning and purpose as the person makes efforts to overcome the effects of SMD [17] and includes concepts such as participation and resilience [18]. It allows individuals to participate in an active life, understand their abilities, generate hope, enjoy personal autonomy, social identity, meaning and purpose in life, and experience a positive sense of self [19,20].

There are models that describe the components of recovery and others that detail the steps for its development. Notable among them are the CHIME Model [21] with five interrelated processes described as connection, hope, identity, meaning, and empowerment; the TILDA Model of Recovery [22]; and the Stages of Recovery Model [23].

From the perspective of people with SMD, research indicates that experiences are overwhelmingly negative. However, aspects that positively influence recovery include clarity around routines and rules, engaging in stimulating activities, meaningful relationships with the healthcare team, contact with nature, and a safe and warm environment [24].

The participation of nurses in the recovery process of people with mental health or psychiatric problems is peripheral to the biomedical model, partly because an approach focused on symptom relief still predominates [25,26] in restrictive environments laden with behaviour control practices [27], which prioritise safety and the mitigation of risks or harm [17]. Administrative tasks are prioritised and brief contact, usually related to specific tasks, is favoured [28]. When measuring knowledge about recovery, nurses score lower than doctors when using scales such as the Recovery Knowledge Inventory (RKI), and there has been little progress in the last 10 years [29].

Several studies support the incorporation of nurses as key agents in mental health promotion and prevention, in addition to the treatment and recovery of people with psychiatric illness [18,30]. Personal recovery can be strengthened through specialised nursing interventions [31,32], focused on connecting with individuals and their families [17], with effects that even reduce episodes of confinement and mobility restrictions [33] or decrease hospital readmissions through empowerment at key moments [34]. Successful examples include Anxiety Communication Notes (ACN), which provided structure in the development and practice of adaptive anxiety management for patients through visual representation for mutual understanding of their anxiety patterns [26], or the Shared Decision-Making (SDM) strategy, through which the nursing team creates opportunities for self-understanding and self-determination in hospitalised individuals [35].

Based on the background presented, the objective of this research was to explore the participation of nurses in the recovery process of people hospitalised in Psychiatric Intensive Care Units (UHCIP by its Spanish acronym) in Chile between 2023 and 2024.

## 2. Materials and Methods

### 2.1. Design

This manuscript was drafted against the International Comitee of Medical Journal Editors (ICMJE) and COREQ_Checklist [36] for qualitative research. The research was conducted under the qualitative paradigm, with a multi-site collective-case study design, which sought to obtain a deep understanding of similar cases from different perspectives [37]. This design is based on inductive descriptive reasoning, whose purpose is to represent a factual reality [38]. In the case study, the researcher explores systems defined over time through detailed and in-depth data collection [37]. Convenience sampling was used to select individuals who offer a rich and in-depth perspective on the phenomenon under study [37]. Participants were selected based on the following criteria: relevance, suitability, appropriateness, timeliness, and availability [38].

### 2.2. Research Procedure and Data Collection

In the context of doctoral studies at a Spanish public university, the research protocol was submitted to the Ethics Committees of four hospitals in the capital of Chile, three of which belong to the public health service. After approval by each committee, an invitation was extended in person to the nurses in each unit. Those who expressed interest were asked to provide their email address so that the Informed Consent (IC) form could be sent to them. Semi-structured individual interviews were arranged with those nurses who returned the signed IC form in order to gain a deeper understanding of the phenomenon under study. Semi-structured individual interviews were arranged with those nurses who returned the signed IC to gain an in-depth understanding of the phenomenon under study [37]. These were conducted in Spanish and online via Zoom^®^ between October 2023 and July 2024. Only the audio was retained, and the video was discarded. The content of the interview script (Table 1) was developed based on different areas of interest according to the available literature and was tested in a pilot focus group, the results of which were not included in this final results analysis. The interviews continued until information saturation or when no new content was obtained [38].

### 2.3. Participants

The research was conducted with nurses working in the aforementioned UHCIPs in the capital of Chile, who met the following inclusion criteria: age 18 or older; professional nursing degree; one year of experience in a UHCIP. Eighteen nurses were included in the study, with a mean age of 32.72 years and 7.36 years of experience in mental health and psychiatry. Of these, 17 were women, 11 belonged to UHCIPs in the public health system, and 7 belonged to the private system. The sociodemographic characteristics of the participants are shown in Table 2.

### 2.4. Data Analysis

A rapid qualitative analysis was performed, a timely and relevant type of analysis that has proven to be useful and fast, compared to other classical methods [39], while maintaining rigorous criteria [40]. This time advantage allows for the exchange of findings [41] and is particularly interesting for health researchers who are interested in having their results become part of protocols, standards, or public policies that impact people’s quality of life [39]. The rapid qualitative analysis method includes preparing the data in a complete and orderly manner in a spreadsheet, familiarising oneself with it through repeated reading, rapid coding with the identification of patterns and categories, identifying representative textual citations, summarising by establishing relationships, and rapid validation by making adjustments [40]. The various approaches to rapid qualitative research, such as Rapid Ethnographic Assessments (REA), Rapid Qualitative Inquiry (RQI), and real-time evaluations (RTE), among others, have been described as having an iterative design, combining multiple data collection methods and allowing for short study periods [39]. This avoids word-for-word transcription and allows interview notes to be generated [42,43]. In general terms, it includes the creation of a neutral domain name for each question, the creation of a data summary template, usability and relevance checks, the allocation of transcripts among the team, the transfer of summaries to a matrix, and the synthesis of results [44,45].

### 2.5. Ethical Considerations

In accordance with the Declaration of Helsinki and the CIOMS 2016 International Ethical Guidelines for Research Involving Human Subjects, the research complies with social value, scientific validity, risk/benefit assessment, declaration of conflicts of interest, use of informed consent, and protection of participants’ rights.

The research protocol was submitted to the following ethics committees (Table 3):

## 3. Results

### 3.1. Description of Nursing Care in Mental Health and Psychiatry

The work of nurses in UHCIP is mainly described as being associated with the care role, carried out by a variable number of people ranging from 11 in the private system to 40 in the public health system. The description provided by nurses includes the development of an individualised nursing care plan according to each person’s situation and direct supervision of risks (self-harm or harm to others, suicide, escape and falls, among others). In addition, they describe the performance of nursing procedures such as taking tests, installing venous accesses, installing nasogastric tubes, administering oral, intravenous and intramuscular medications, and performing dressings. From a management perspective, they describe the supervision of technical staff, management of supplies, consultations, monthly statistics, and compliance with institutional accreditation standards. From an educational perspective, the nurses interviewed refer to different scenarios. On the one hand, they mention the limited time they have to educate hospitalised patients and their families, for example, about preparing for discharge home, as seen in the following quote: “*I feel like I’m not contributing anything… because look, I have… I don’t know, 40 patients under my care, and out of those 40, I manage to follow up on 4–5… That’s why I was saying… to have formal psychoeducation sessions where you can take the person, take the family*” E08(17:53).

On the other hand, other interviewees mention the successful experience they have had within the team implementing education groups for individuals and family members that continue after discharge, to which they attribute a fundamental role in recovery, as can be seen in the following quotes: “*The nurse does a lot of mental health education, which is carried out… and (so that) patients do not ultimately relapse and do not have to return to a process of rehospitalisation*” E04(04:05); “*With education, I am not going to make them recover from bipolar disorder, but I am going to make it possible to reduce the recurrence of the disease*” E05(04:55). Finally, none of the interviews mentioned the role of research in nursing.

### 3.2. Relationship Between Nursing and Recovery: Contributions from the Discipline

Most of the nurses interviewed link nursing with recovery through the genuine therapeutic relationship established with hospitalised patients. In the context of general treatment, it is thanks to the quality of the bond established that people manage to turn their situation of psychological vulnerability around and can glimpse aspects of their lives that improve their experience of illness. For these nurses, the bond is part of a complex process that is built over time and involves elements such as valuing the person’s autonomy, trusting in their abilities, and creating spaces for shared decision-making, as shown in the following quote: “*From the nursing role… we inquire and determine what the patient wants to emphasise, their aspirations upon discharge, their willingness to move forward*” E01(08:48).

Among the therapeutic tools that strengthen the bond and promote recovery, nurses strongly mention the development of mutual trust and active listening. They report that people hospitalised in a UHCIP continually express hopelessness, have lost the ability to visualise positive aspects of their lives, neglect their personal self-care, or fail to visualise the support networks available to them, all of which make the hospitalisation space a challenge for the nursing team. When nurses become a resource, thanks to the trust they have built, spaces for dialogue are fostered in which different coping strategies or alternative solutions that the person had not previously been able to visualise are explored: “*…So when she asks for help, she is willing to try certain tools… over time, it becomes a team effort… you can see a change, as she tries to manage this and use these skills… and we saw progress…*” E18(05:26). Active listening requires the nursing team to stay focused, always returning to the interests and challenges that the hospitalised person themselves has set for their recovery. In the context of this therapeutic relationship, active listening encourages the nursing team to focus on what people really want, revealing their interests, as shown in the following quotes: “*I immediately thought about the bond… we see them 12 h straight, sometimes 24 h straight, where you really get to know them intimately, in their daily lives… The trust that is built with the nursing staff goes hand in hand with confidence, and that confidence translates into decision-making.*” E05(10:11); “*A mother who has young children who cannot enter the unit… that patient may be at high risk of escape, but she is asking to see her baby, so we set up a safe environment, so that the patient does not escape, but can see her baby.*” E8(07:48).

The interviews also mention other elements that contribute to the recovery of people in a UHCIP, described as a safe physical space, compliance with rules and protocols, sufficient nursing staff, management of material resources, and availability of medicines, among others.

### 3.3. Components of Recovery

The vast majority of nurses interviewed agree that the components of a person’s recovery in a UHCIP are related to four elements, defined as the existence of family and social support, the ability to resume control of one’s own life around what one values, the existence of a life plan for the future, and the development of the ability to manage one’s own condition. With regard to family and social support, they mention that its existence and effective functioning contribute enormously to therapeutic success, as it acts as an early warning sign of decompensation or supports aspects of treatment upon discharge: “*…there are patients who have a large support network, whose family is present… you can see that there is better follow-up… some are working or have resumed their studies, but of course, they need that support from someone who is with them, who helps them with everything, the medication or check-ups, to know what the warning signs are, why they should consult*” E09(11:59). The interviewees also agree that people without family and social support tend to have slow recovery processes with multiple relapses, which chronifies psychiatric conditions, increases rehospitalisations and tends to impoverish people.

With regard to the ability to lead one’s own life according to one’s values, the interviewees mention the importance of focusing nursing care on what the person wants for themselves. Some of them agree that this action represents a substantial transformation of the usual biomedical model, as it requires a “change” to establish therapeutic goals focused on what the person wants, which is an unusual practice in mental health and psychiatric teams, accustomed to making all the decisions, as can be seen in the following quote: “*it has to do with the person’s goals…so that they can then be self-sufficient… it has a much broader impact than if only the doctor is responsible for prescribing the drugs and the rest are responsible for delivering them… the person is not understood in this way, but as having a lot of facets and requirements that go beyond the merely biomedical or pharmacological.*” E13(12:26).

Some interviewees mention that developing the ability to manage one’s own condition is a process that does not necessarily culminate in UHCIP and that it depends on several elements, among which the acceptance of the condition (and the grief that this implies) and the development of concrete actions aimed at self-care stand out. Acceptance of the condition is what would lead to the timely management of symptoms, requesting help from the nursing team when the person considers it necessary. With regard to self-care, all interviewees mentioned the importance of supporting people in a UHCIP to regain their level of autonomy in areas such as eating, elimination, hygiene, sleep and rest, recreation, and hazard prevention, among others. They highlight the caregiving role of nursing, as the professionals who par excellence promote self-care and ensure that learning in this area is maintained over time, until the person regains their level of self-sufficiency: “*…which is also the normalisation of certain biological cycles such as sleeping, eating, elimination… these are cycles that are sometimes altered by mental health pathologies, therefore, in recovery one also sees the normalisation of these cycles in people.*” E11(12:29).

Finally, in this domain, many interviewees agree that having a life plan for the future makes a difference at the time of discharge, becoming a driving force for personal recovery. Various members of the team are involved in the development of this plan, with nursing staff collaborating in a less active role. The work of occupational therapists and the role they play in this area is strongly mentioned, as they explore areas of student, occupational and work interests once the person is close to discharge, as is the work of psychologists with family members to develop plans for the person’s future.

### 3.4. Definition of Recovery

All interviewees agreed that recovery is a concept that adapts and varies according to the characteristics of each person hospitalised in a UHCIP. Depending on the person’s interests, recovery may focus on different components. However, all agree that recovery translates into the ability to resume the life that was diminished or lost during the illness and to regain functionality. It involves everyday aspects such as managing money again, enjoying a trip to the cinema, performing physical activity or enjoying one’s work. Some interviewees mentioned that the experience of psychiatric hospitalisation is similar to a storm, so recovery would translate into the ability to remain stable after being affected, even coming out of the experience stronger (similar to the concept of resilience), as can be seen in the following quote: “*…it’s like stepping out of the storm that brought you here, but the hardest part is leaving here, being able to remain stable in your environment, because here you are in a super-protected environment, free from harmful stimuli.*”E10(13:57).

Other interviewees mention measurable or observable elements of recovery from the perspective of professionals, such as the person understanding their diagnosis, achieving symptom stabilisation, adhering to medication, attending check-ups, etc. These can usually be assessed using scales or questionnaires, but if they do not reflect the real needs or interests of the individuals, they do not promote true recovery, as can be seen in the following quote: “*But it often happens that the team in general tries to look for this… but I feel that it makes less sense to the person when others make these decisions for them…*” E16(15:57). In the same vein, some professionals also refer to the remission of the symptoms for which the person was admitted, improvement in their altered judgement of reality, absence of suicide risk or effective detoxification.

### 3.5. Use of Instruments in the Assessment of Recovery

A very small number of nurses interviewed recommend the use of self-administered instruments in the assessment of the recovery of a person hospitalised in a UHCIP. They cite as reasons the lack of time and having to devote this space to other tasks. It is argued in this regard that the decompensation of individuals does not allow them to focus their attention on an instrument or respond to it coherently, which would eventually change as discharge approaches, but that this is not performed either due to the high workload. On the other hand, there was no agreement among them regarding the usefulness of using instruments at different times during hospitalisation. Some mentioned that it would be ideal to apply them upon admission and discharge, while others would apply them only when medical discharge is approaching. Regarding the nurses who did recommend their use, they argued that it would possibly be an opportunity to give voice to the wishes of the hospitalised person and to work on those aspects that are relevant or that the person wants to prioritise, as exemplified in the following quote: “*It is important to apply it a little before discharge to see what [the hospitalised person] thinks, because there are many elements that help us investigate things beforehand, or how confident they are about being able to reintegrate…*” E17(26:44). Other reasons given for the impossibility of using self-administered measurement scales refer to a lack of coordination with the healthcare team and a lack of knowledge about their usefulness.

## 4. Discussion

Nursing care in mental health and psychiatry makes sense when the person progresses in their recovery process, which often begins with a stay in a UHCIP. In this regard, the results of the interview analysis coincide with the elements described in contemporary approaches to recovery [13,14,18,20], incorporating aspects such as autonomy, trust in the person’s abilities, and shared decision-making [15]. Particularly noteworthy are the accounts associated with the therapeutic relationship of closeness and mutual respect that allow for a real bond with the person and their family [17], in which spaces for dialogue are established to work on new coping strategies. As reported in other research [24], the nurses interviewed assigned an important role to contextual elements such as safe space, management of material resources, and compliance with rules and protocols. Close to this analysis is the consideration that recovery depends on the satisfaction of basic needs and work on self-care, similar to more classical conceptualisations of clinical recovery incorporated, for example, in Henderson’s nursing models with her 14 basic, psychosocial, and spiritual needs, and Orem’s with her universal self-care requirements, developmental requirements, and health deviation requirements [46].

Regarding the components of recovery, the interviewees refer to family and social support [14], leading one’s own life [20], the existence of a life project [19], and the ability to manage one’s own condition [17]. If these components are analysed from the CHIME Reference Framework [21], the results of the interviews largely coincide with those of recent research [47], which has shown, for example, that working on hope is the best predictor of overall changes in the psychological recovery of people diagnosed with schizophrenia [48]. Nurses also mention that the absence of family and social support leads to recovery processes with multiple relapses that delay reintegration [4], which makes psychiatric conditions chronic and prolongs treatment. This results in increased rehospitalisations [8] at a high cost that tends to impoverish people [9], which is a constant feature in the mental health and psychiatric contexts of a country that has only had a Mental Health Law since 2021 [2]. Unfortunately, in relation to promoting self-directed living and the ability to manage one’s own condition, most of the nurses interviewed continue to focus their work on classic approaches that seek to control behaviour [27], symptom relief [25,26] and risk or harm mitigation [17], which is not consistent with current approaches in the field that promote self-determination in hospitalised individuals [35]. On the other hand, brief contact, usually related to specific tasks, is favoured [30], which relegates nursing work in mental health and psychiatry once again to meeting basic needs or administrative procedures.

Although the nurses interviewed recognise that the person’s recovery is the goal, the interviews did not reveal a consensus on its meaning and purpose [17], but rather a disaggregated description of elements involving its components. It was not possible to identify the existence of a recovery-centred care model, so there is a significant gap in the meaning of nursing care to facilitate an active life for the person, in which hope grows, or in which they can enjoy personal autonomy with a positive sense of self [19,20]. Finally, no consensus was observed on the usefulness of specialised recovery assessment tools, which increase personal autonomy [26] and self-understanding and self-determination in hospitalised individuals [35], a fact that may possibly be related to the high workload faced by professionals in their daily work [49].

Implications for Clinical and Academic Practice:Mental health and psychiatric nurses are experts in care throughout the life cycle.Redirecting this care towards recovery is a challenge that requires a paradigm shift towards a model in which the person affected by a mental health condition is the protagonist of their own health process.Specialised nursing teams are responsible for ensuring respect for mental health rights that are often violated.

## 5. Conclusions

This qualitative research explored the participation of nurses in the recovery process of patients hospitalised in a UHCIP in Chile, which contributes to the discipline in the context of the current process of recognition of the specialty in Chile. From the nurses interviewed, it can be concluded that recovery is a dynamic process that is unique to each individual, influenced by factors such as the drive to live life to the fullest, the existence of a life plan, the ability to manage one’s own illness and its symptoms, and the existence of family and social support, all of which are described in emerging models such as the CHIME Model. It is revealed that mental health and psychiatric nursing and specialised care in this area play a central role in the recovery process of people affected by a condition, through the articulating axis of the therapeutic relationship based on active listening, respect for people’s autonomy, and trust. Elements of the therapeutic relationship and bond are highlighted as drivers of personal recovery, associated with components that the interviewees clearly identify as the existence of a future life plan, the ability to manage their own condition, family and social support, and autonomous control of their own lives. Despite this, the recovery approach used by nurses is still classic and exclusively clinical, assessable only from the perspective of professionals, which is far from new approaches to personal recovery. Based on these results, it is essential to highlight the central role of the individual in managing their own recovery process, incorporating the family and the community into an active process that requires constant adjustments.

The qualitative methodological design is a strength of the study, as it allows for a deeper understanding of the experiences of the nurses interviewed through rapid qualitative analysis of each individual interview. These results will contribute to the current discussion on the recognition of the specialty, requiring legislators and decision-makers to incorporate work on recovery from primary healthcare across the entire network. One of the limitations of this research relates to the number of subjects involved and the fact that the perspective of hospitalised patients was not included.

According to the evidence reviewed and the results obtained, it is essential to change the nursing care paradigm if we want to work with people around recovery. It would be interesting to continue exploring this topic in future research that lays the foundations for the construction of a nursing care model for recovery that is culturally adapted to the local reality and that considers the historical evolution of the discipline, as well as developing effective strategies for the implementation of recovery-oriented practices in UHCIPs, for example, using specialised recovery instruments.

## Figures and Tables

**Table 1 nursrep-15-00391-t001:** Semi-structured question script.

1.	What does your work in this unit consist of?
2.	What actions in your daily work are aimed at the recovery of the people you care for?
3.	What elements determine a person’s recovery from your perspective?
4.	How does nursing work contribute to the recovery of people discharged from this unit?
5.	When would you say a person is recovered?
6.	In what ways could the recovery perspective be incorporated into your daily work?

**Table 2 nursrep-15-00391-t002:** Sociodemographic characteristics of participants.

Participant (P)	Gender	Age (Years)	Experience at UHCIP (Years)	Current Working UHCIP	Interview (Minutes)
P1	Female	26	1.5	Clínica Universidad Los Andes	34:17
P2	Female	30	7	Hospital Dr. Sótero del Río	26:09
P3	Female	33	6	Instituto Psiquiátrico Dr. José Horwitz	27:27
P4	Female	32	10	Hospital de la Florida Dr. Eloísa Díaz	26:57
P5	Female	33	10	Clínica Universidad Los Andes	28:26
P6	Male	31	7	Hospital de la Florida Dr. Eloísa Díaz	28:25
P7	Female	27	4	Instituto Psiquiátrico Dr. José Horwitz	38:23
P8	Female	44	11	Clínica Universidad Los Andes	32:55
P9	Female	29	2	Instituto Psiquiátrico Dr. José Horwitz	24:28
P10	Female	34	6	Clínica Universidad Los Andes	33:28
P11	Female	41	15	Hospital Dr. Sótero del Río	38:52
P12	Female	44	20	Instituto Psiquiátrico Dr. José Horwitz	37:58
P13	Female	31	9	Hospital de la Florida Dr. Eloísa Díaz	34:46
P14	Female	35	10	Clínica Universidad Los Andes	36:52
P15	Female	34	4	Clínica Universidad Los Andes	33:15
P16	Female	30	8	Hospital de la Florida Dr. Eloísa Díaz	27:07
P17	Female	26	1	Hospital de la Florida Dr. Eloísa Díaz	29:22
P18	Female	29	1	Clínica Universidad Los Andes	26:20

**Table 3 nursrep-15-00391-t003:** Ethics committees that approved the research.

	Name of the Committee	Date of Approval
1.	Scientific Ethical Committee of the Hospital Clínico La Florida Dra. Eloísa Díaz Insunza (CEC-HLF), belonging to the Servicio de Salud Metropolitano Sur Oriente.	13 May 2021
2.	Research Ethics Committee of the North Metropolitan Health Service (CEI-SSM)	18 May 2022
3.	Scientific Ethical Committee of the Universidad de Los Andes Clinic (CEC-CUA)	12 September 2022
4.	Scientific Ethical Committee of the Southeast Metropolitan Health Service for Dr. Sótero del Río Hospital (CEC-SSMSO)	4 May 2023

## Data Availability

Research data are available to anyone who wishes to request them and can be obtained by writing to the author’s email address.

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
