# Peer review of "Nurses’ Participation in the Psychiatric Recovery Process: A Qualitative Study in Psychiatric Intensive Care Units in Chile"

_nursrep, 2025, doi:10.3390/nursrep15110391_

Round 1

Reviewer 1 Report

Comments and Suggestions for Authors
  • The introduction needs improvement due to the very weak of connection to the main
    title of the manuscript;
  • no information on how many hospitals out of four accepted the research protocol submitted to the Ethics Committees of the hospitals in the capital of Chile;
  • too small a research group 18 cannot provide statistical reliability of the study.
    Therefore, it is recommended to change the title from "Nurses’ participation in
    the psychiatric recovery process " to "Preliminary studies:Nurses’ participation in
    the psychiatric recovery process”;
  • section called "Implications for Clinical and Academic Practice" would be welcome
    prior the ,,Conclusion” paragraph, as it would increase value of this work.
    The recommended paragraph would be more readable in the form of bullet points
    rather than as a single text.The implications should include indisputable messages
    resulting from this research.

Author Response

Thank you very much for taking the time to review my work. Please see the attachment.

Reviewer 2 Report

Comments and Suggestions for Authors

Please see attached annotated draft. Thank you.

Author Response

Thank you very much for taking the time to review my work. Please see the attached file.

Reviewer 3 Report

Comments and Suggestions for Authors

This study contributes to the recognition process for nursing psychiatric specialties in and is is innovative in that it explores nurses' participation in the recovery process of individuals hospitalized in   Psychiatric Intensive Care Units. I consider it very thoroughly designed and prepared. 

Definetely I would reconsider the title change as it is imprecise so adding a subtitle may help eg. 

Nurses’ participation in the psychiatric recovery process: a qualitative study among nurses working in psychiatric intesive care units in Chile

I consider the qualitative methodological design as a strenght of the study as it allows for a deeper understanding of the experiences of nurses, rather than a survey of sorts. Rapid qualitative analysis was used for each individual interview i would think about Interpretative Phenomenological Analysis as an additional method? Maybe. I leave it to Authors. 

The study results provide information that will lay the foundation for future research on building a recovery-focused nursing care model that is culturally adapted to the local reality, which is sadly to say yet being focused on the role of the nurse rather as "programmed robot" to do certain ordered tasks, so definetely its findings should be disseminated among eg. management personel of such psychiatric care units.

Author Response

(The authors gave the same response as above.)

Round 2

Reviewer 1 Report

Comments and Suggestions for Authors

The authors took into account the reviewer's recommendations and improved the text in satisfactory manner.